# Description and Analysis of Glycosidic Residues in the Largest Open Natural Products Database

**DOI:** 10.3390/biom11040486

**Published:** 2021-03-24

**Authors:** Jonas Schaub, Achim Zielesny, Christoph Steinbeck, Maria Sorokina

**Affiliations:** 1Institute for Inorganic and Analytical Chemistry, Friedrich-Schiller University, Lessing Strasse 8, 07743 Jena, Germany; jonas.schaub@uni-jena.de; 2Institute for Bioinformatics and Chemoinformatics, Westphalian University of Applied Sciences, August-Schmidt-Ring 10, 45665 Recklinghausen, Germany; achim.zielesny@w-hs.de

**Keywords:** natural products, glycosides, bioactivity, glycosidic residues, sugars, carbohydrates, deglycosylation, cheminformatics, Chemistry Development Kit, CDK

## Abstract

Natural products (NPs), biomolecules produced by living organisms, inspire the pharmaceutical industry and research due to their structural characteristics and the substituents from which they derive their activities. Glycosidic residues are frequently present in NP structures and have particular pharmacokinetic and pharmacodynamic importance as they improve their solubility and are often involved in molecular transport, target specificity, ligand–target interactions, and receptor binding. The COlleCtion of Open Natural prodUcTs (COCONUT) is currently the largest open database of NPs, and therefore a suitable starting point for the detection and analysis of the diversity of glycosidic residues in NPs. In this work, we report and describe the presence of circular, linear, terminal, and non-terminal glycosidic units in NPs, together with their importance in drug discovery.

## 1. Introduction

Natural products (NPs) are biologically active molecules produced by living organisms. Their importance for the producing organisms themselves and for their interaction with their environment as well as for therapeutical and agricultural usages is widely accepted [1,2,3]. Many NPs are glycosides, which means that in addition to their main structure (the aglycon), they contain one or several glycosidic substructures. For the glycosylated NP, the sugar residues can be crucial for their bioactivity, and when it is dispensable, they improve the pharmacokinetic parameters of the molecule [4]. Important classes of natural compounds are glycosides, such as hormones, alkaloids, polyketides, flavonoids, or antibiotics.

Due to the hydrophilic structures of their sugar moieties, glycosides tend to be more water-soluble than their aglycons [4], which influences their pharmacokinetic properties, such as their concentration, circulation, and elimination in the human body fluids. However, this increased hydrophilicity mainly influences negatively the transport of the molecules through membranes, as hydrophobic compounds tend to enter cells due to their solubility in lipids. On the other hand, some glycosides can be actively transported into tissues and cells using the glucose transport system.

Streptomycin (Figure 1a), an antibiotic isolated in 1943 by Albert Schatz from the bacterium *Streptomyces griseus* and still used in the present day, is a good example of the impact of a glycoside on the bioactivity of the molecule. The amino groups of the sugar rings of this antibiotic class bind to ribosomal RNA, causing a decrease in translation and translational accuracy and inhibit the translocation of the ribosome [5,6]. Aureolic acids are a group of antibiotics with one or several particular sugar moieties, D-olivose, attached to the aglycon (in Figure 1b, olivomycin A, a member of the aureolic acid family) and are known to inhibit the DNA-dependent RNA polymerase. Members of this group that are lacking some of the sugar moieties are less active [7], and the aglycone, shared by all the group members, is not active as it does not bind to the DNA at all [8]. Sugar moieties are also receiving attention in NP-derived antibiotics, such as vancomycin, a last-resort antibiotic where the race to avoid antibiotic resistance is focusing on modifying the glycosidic residues which are targeted by antibiotic resistance genes [9].

Surprisingly, glycosylation in vitamins often reduces their bioavailability [4]. For example, the natural variant of pyridoxine (Figure 1c), commonly known as vitamin B6, is significantly less bioavailable. The uptake of vitamin B6 can decrease by 70–80% when the glycosidic residue is present [10,11].

Cardiac glycosides (Figure 1d), a well-known family of NPs from plants, are the so-called “cardiac steroids” as some of them are structurally similar to the adrenal cortical hormones in higher mammals and potentially use the same binding site. They inhibit Na+/K+-ATPases, which is useful for the treatment of several heart conditions [12]. This activity is enhanced several-fold with the presence of sugar moieties in these plant-extracted compounds [13].

Extracts from the root of *Panax ginseng* have been used in traditional medicines for more than 5000 years [14]. Modern approaches identified a number of bioactive NPs, in particular ginsengosides (Figure 1e), which are dammarane-type saponins. Compounds from this chemical class are highly glycosylated, and their bioactivity often differs by the number of sugar moieties attached, varying from central nervous system stimulation to the increase in gastrointestinal motility and weak anti-inflammatory action [15,16].

The very few examples presented here are only an attempt to illustrate the diversity of glycosidic moieties across the different domains of life and their importance in the bioactivity and bioavailability of NPs. To advance the understanding of their mode of action, it is therefore important to correctly detect the glycosylated NPs in order to prioritise them for drug discovery and other health and agriculture applications. In the present work, we analysed and described the compound glycosylation in COCONUT [17]. COCONUT is currently the largest collection of elucidated NPs and predicted NPs from various taxonomic origins that was assembled from more than 50 diverse open NP sources. Our aim is to provide an easy and detailed overview of the occurrence of sugar moieties in a very large NP set, together with their structural properties.

## 2. Methods

### 2.1. Detection of the Glycosidic Moieties

For the detection and removal of glycosidic moieties in the molecular datasets, the open-source software Sugar Removal Utility [18] (SRU) was used. Based on the Chemistry Development Kit [19] (CDK), it consists of a generalised and configurable approach for in silico detection and removal of circular and linear sugar moieties from chemical structures.

In the present study, the SRU default options were used: only terminal sugars were removed, non-glycosidic side chains connected to sugar moieties were preserved if they had 5 or more heavy atoms, circular sugars with a ratio of exocyclic oxygen atoms to atoms in the ring of more than 0.5 were detected. A sugar moiety is terminal when its removal does not split the aglycon into two or more disconnected substructures. Additionally, spiro rings, rings with keto groups, and linear sugars in rings were excluded from detection by the SRU. Only non-acidic linear sugars whose sizes are between 4 and 7 carbon atoms were detected. However, for some of the analyses, as outlined in the Results section, non-default options were used, e.g., to distinguish between terminal and non-terminal sugar moieties.

The prevalence and characteristics of glycosidic residues in several molecular collections were investigated using the SRU: in the COlleCtion of Open Natural prodUcTs (COCONUT) and the ZINC “for sale” and “in vitro” subsets.

### 2.2. COCONUT

The COCONUT database [17] (available online: https://coconut.naturalproducts.net, accessed on 1 October 2020) contains 401,624 predicted and confirmed NPs and is currently the largest open molecular collection of this type. It has been assembled from 53 open NP resources and contains structural, physicochemical, taxonomic, and geographic data for NPs of diverse origins, including plants, bacteria, fungi, and animals. This dataset is therefore the optimal choice for the exploration and statistical description of the glycosylation in NP.

### 2.3. ZINC

ZINC15 (available online: http://zinc15.docking.org, accessed on 1 October 2020) is an open database of small commercially available molecules that has been initially developed for virtual screening. Molecules in ZINC15 are organised in various subsets, some of which are based on their natural or synthetic provenance. Therefore, ZINC15 is an excellent resource for retrieving sets of structurally diverse synthetic molecules. In order to compile such a diverse dataset of synthetic molecules to compare their glycosylation with the NPs in COCONUT, 22 Mio molecules of the ZINC15 [20] “for sale” subset were downloaded and curated to ensure that the subset did not contain any NPs nor primary metabolites (Figure 2a). As a first step, the dataset was reduced in size to 500,000 molecules using the RDKit MaxMin algorithm [21,22] to preserve diversity. From the reduced set, all molecules were removed that matched with molecules in the ZINC15 “biogenic” subset, which contains primary and secondary metabolites, or with COCONUT molecules. The structure matching and filtering were performed without regarding stereochemistry, i.e., stereoisomers of primary metabolites or NPs were filtered even if they were not annotated as such in ZINC. Additionally, stereoisomers of the remaining synthetic molecules were grouped to avoid redundancy. This way, a dataset of 475,958 diverse synthetic molecules was obtained, similar in size to COCONUT (401,624 molecules). Matching was performed using unique SMILES representations of the molecules generated by the CDK.

The ZINC15 “in vitro” subset was also downloaded and curated in the same way. It contains molecular structures that have shown activity in in vitro binding assays. The purpose of the deglycosylation of this dataset was to compare the glycosylation statistics of NPs and synthetic molecules with active synthetic molecules. After removing all biogenic molecules from the original “in vitro” subset and grouping the stereoisomers of the remaining compounds, its size was reduced from 306,346 to 65,620 molecules.

### 2.4. Sugar Moieties in Bacterial Natural Products

While the present study focuses on extracting and analysing sugars from NPs using a computational tool, Elshahawi et al. previously published a review on manually curated glycosidic moieties in bacterial NPs containing 344 distinct cyclic glycosides [23]. In COCONUT, a substructure search for these moieties was carried out to analyse their occurrence and distribution, additionally checking whether they are detectable by the SRU. The 344 reported glycosides were first grouped by stereoisomers using unique SMILES representations generated by the CDK (Figure 2b). Then, they were separated in two datasets, depending on whether they are detectable with the SRU default options as sugar moieties or not. Finally, a substructure search for every stereoisomer group of both datasets was performed in COCONUT using the CDK class *DfPattern,* and the matches counted. However, the number of substructure matches detected here would not precisely correspond to the number of cases where the SRU would remove the glycosidic moiety as the SRU does not only detect the presence of a substructure but also detects whether it is terminal or an isolated cycle.

All analyses of the glycosides for the used compound datasets were performed with Java 11. The code is available on GitHub https://github.com/JonasSchaub/GlycosylationStatistics, accessed on 1 November 2020, along with the scripts used for the curation of the ZINC datasets.

## 3. Results

### 3.1. Deglycosylation and Glycoside Analysis in COCONUT

#### 3.1.1. Description of Sugar Moieties in COCONUT

Table 1 summarises the absolute numbers and respective proportions of different types of glycosidic NPs detected in COCONUT with the SRU. The October 2020 version of COCONUT contains 401,624 unique NP structures. In this dataset, the SRU detected 11.98% (48,118) NPs with at least one sugar in their structure. A total of 97.63% (46,979) of NPs with sugars contain circular sugars, and 96.41% (46,393) contain only circular sugars, no linear ones. An example for the latter is 6-C-lucopyranosylpilloin depicted in Figure 3a. If the respective SRU option is activated, 697 NPs containing circular sugars in spiro-form can be detected, e.g., moriniafungin F (Figure 3b). Only 3.67% (1725) of glycosylated NPs from COCONUT contain linear sugars, and 66.03% (1139) of these have no circular glycosidic moieties attached to the parent structure, only linear ones. The sesquiterpene glycoside cosmosporaside A (Figure 3c) is a representative of this group. In the whole dataset, only 586 NPs have both circular and linear sugars decorating the parent structure, as it is the case for llvaudioside B (Figure 3d).

A total of 1404 NPs in COCONUT are composed only of sugar units with side chains or aglycons of less than five heavy atoms, so they are removed completely by the SRU. In Figure 4, multiple examples are compiled to illustrate this class of NPs that can be further subdivided: 30.27% (425) of these NPs are single sugar units (circular in 50.82% (216) and linear in 49.18% (209) of cases, Figure 4a–d), and the remaining 69.73% (979) are carbohydrate polymers (Figure 4e–g). A total of 93 molecules in COCONUT are circular sugar monomers without a glycosidic bond, which represent a special case in SRU circular sugar detection.

The SRU distinguishes between terminal sugar units, which can be removed from a molecule without creating a disjointed structure as a remaining aglycone (Figure 5a) and non-terminal ones where this is not the case (Figure 5b). Regarding this distinction in COCONUT, 74.86% (35,167) of the circular sugar-containing molecules in the dataset have only terminal circular glycosidic moieties and no non-terminal (circular) ones, similar to the cardiac glycoside diginatin (Figure 5a). On the other hand, 7762 NPs contain only non-terminal circular sugar moieties (Figure 5b), and even fewer molecules (4050) contain both terminal and non-terminal circular glycosidic moieties (Figure 5c).

Regarding the linear sugars, 1117 molecules contain only terminal linear sugars and zero non-terminal linear ones (64.75% of linear sugar-containing molecules, Figure 6a), in contrast to 606 (35.13%) molecules for the opposite case. The latter can be illustrated by aspergillusol A depicted in Figure 6b that has a non-terminal tetraol moiety and no additional terminal linear sugar moieties. Only two NPs in COCONUT have both terminal and non-terminal linear glycosidic moieties in their structure. One of them is the COCONUT NP CNP0321545 depicted in Figure 6c.

A total of 49.42% (23,780) of sugar-containing NPs have only one sugar moiety in their structure (Figure 7), and the vast majority (96.38%, 46,377 molecules in total) have five or less sugar moieties, such as dalmaisiose P (Figure 8a). However, there are some exceptional cases, e.g., four NPs that each have 13 sugar moieties and three molecules having 14. One of them is glycan G00008 given in Figure 8c. In these three NPs, all sugar moieties are circular. The molecule with the most linear moieties detected is depicted in Figure 8b and has four terminal linear glycosidic moieties.

The previous sections described the glycosylation in COCONUT based on the molecules carrying sugar moieties, e.g., how many molecules have circular or linear sugars. In the following, the focus shifts from the whole molecules to only their glycosidic moieties that are detected and can also be removed or extracted by the SRU.

There is a total of 99,943 sugar moieties among all molecules in the COCONUT database. The proportions of different types of glycosidic moieties detected with the SRU in COCONUT are summarised in Table 2. Unsurprisingly, the biggest majority of them are terminal ring sugars with an O-glycosidic bond. In detail, there are 98,189 (98.24%) circular sugar moieties, 80,927 (80.97%) terminal circular sugar moieties, and 78,755 (78.80%) terminal circular sugar moieties with a glycosidic bond. Only 1754 linear sugar moieties are detected and out of these, 1139 are terminal. In total, 82,066 terminal and 17,877 non-terminal glycosidic residues are detected, disregarding the circular-linear distinction.

In total, 95.89% (94,149) of the circular sugars are pyranoses, 4.11% (4036) are furanoses, and only four heptoses have been detected in COCONUT. The size distribution of linear sugar residues is quite different: 39.11% (686) of them have four skeletal carbons, 18.36% (322) have five carbons, 40.99% (719) have six carbons, and only 1.54% (27) have seven carbons.

With the corresponding SRU option turned on, 11,950 linear sugar moieties that are found as ring substructures, are detected in COCONUT. Compared to the number of detected linear moieties using the default settings (i.e., ignore linear sugar-like structures in cycles), this number is high. For this reason, a visual inspection of a sample of these cases was performed. The latter revealed that COCONUT molecules with detected linear sugars in rings that are detected by the SRU are very diverse. They include, among others, pseudosugars (sugar-like cycles without ring oxygen, Figure 9a), fused ring systems of sugar-like structures (Figure 9b), macrocycles (Figure 9c), ring systems with an excessive number of hydroxy or keto groups (Figure 9d), and rings that are not included in the circular sugar detection due to double bonds within the ring structure (Figure 9e). This attempted classification along with the example structures in Figure 9 are only supposed to give an approximate overview of moieties detectable with the named SRU option.

One important setting for circular sugar detection by the SRU is the exocyclic oxygen atoms to atoms in ring ratio *r_exo_* threshold. To be detected as such, all potential cyclic sugar moieties should have a sufficient number of oxygen atoms connected to the central furanose/pyranose/heptose ring. This sufficiency threshold is defined as a ratio *r_exo_* of the number of oxygen atoms connected directly to the ring *n_exo_* divided by the total number of heavy atoms (carbon and oxygen) in the ring *n_inc_*:(1)$$r_{exo}=\frac{n_{exo}}{n_{inc}}$$

For example, to reach the default threshold of *r_exo_* ≥ 0.5, a pyranose (*n_inc_* = 6) or furanose ring (*n_inc_* = 5) needs to have at least three exocyclic oxygen atoms (*n_exo_* ≥ 3). As stated above, 98,189 circular sugar moieties could be detected in COCONUT, with the default SRU threshold of *r_exo_* ≥ 0.5. These moieties are represented by all the green bars in Figure 10 combined. The figure shows the frequencies for intervals of exocyclic oxygen atom ratios *r_exo_* of cyclic sugar and sugar-like moieties in COCONUT. A total of 87.30% (85,721) of detected sugar moieties reaching a threshold of 0.5 have an exocyclic oxygen ratio between 0.6 and 0.7, i.e., three oxygen atoms connected to a furanose ring and four to a pyranose ring (Figure 11a,b). An important aspect for interpretation of Figure 10 is that many of the intervals represent only one type of furanoses, pyranoses, and heptoses. For example, the interval [0.5, 0.6) contains mainly pyranoses with three attached oxygen atoms and heptoses having four. However, the latter are very infrequent in general, as previously demonstrated. A furanose cannot have an exocyclic oxygen ratio in this interval. This contributes to the significant height of the bar at interval [0.6, 0.7), because this interval contains both furanoses and pyranoses. The interval [0.7, 0.8) on the other hand contains only heptoses with five exocyclic oxygen atoms attached. Note that when the ratio is 1.0 or higher, some carbons in the ring have more than one oxygen attached to them (Figure 11c,d). While the interval [1.0, 1.1) contains all sizes of sugar rings, all moieties in the interval (1.3, 1.4) are heptoses, exclusively. Lowering the threshold setting to *r_exo_* ≥ 0.4 would detect 2312 additional circular sugar moieties (furanoses and heptoses, Figure 11e), and lowering it to 0 (i.e., completely disregarding the number of attached oxygen atoms to the cycle in the detected moieties) would detect 15,658 additional sugar-like units, of which 32.06% (5020) have no attached oxygen atom at all (Figure 11h).

#### 3.1.2. Bacterial NP Sugar Moieties

To assess the suitability of the SRU to detect and remove frequent, repetitive sugar moieties while preserving structurally important information by neglecting rare glycosides, a manually curated dataset of glycosidic moieties appearing in bacterial NPs [23] was analysed. In order to add a statement about how frequent the respective moieties actually appear in NPs, they were identified in COCONUT using substructure searching. In the first step, all stereoisomers in the dataset were grouped, leaving out 181 individual structures of the 344 from the original set. A total of 86 of the unique sugar moieties were detected by the SRU and 95 were not. Figure 12 shows the frequencies of all analysed sugar moieties that are detected at least ten times in COCONUT, categorised as “SRU positive moieties” (blue bars) and “SRU negative moieties” (orange bars) if they could or, respectively, could not be detected by the SRU with the default settings. Altogether, the glycosides detectable by the SRU are more frequent, although there are also some very frequent moieties that the SRU does not detect. Figure 13 and Figure 14 depict the molecular structures of the glycosidic moieties from the used dataset that were found in COCONUT. The structures in Figure 14 show that 23 of the 41 listed sugar moieties undetected by the SRU can be detected if the exocyclic oxygen ratio threshold is lowered to *r_exo_* ≥ 0.3, i.e., a pyranose ring needs to have at least two exocyclic oxygen atoms attached, including the three most frequent structures. For comparison, the above analysis was re-performed with this parameter altered as described. This way, 120 of the grouped sugar moieties can be detected, and only 61 cannot. Detailed results are shown in Figure 15 and Figure 16. The most frequent undetectable moiety appears about 1900 times in COCONUT and can be made detectable if the exocyclic oxygen ratio setting is lowered further to 0.1. In this case, the most frequent undetectable moiety (the fourth structure in Figure 16) can only be detected if the SRU option to allow the detection of sugar candidate structures having keto groups is turned on. It appears only about 200 times in COCONUT.

### 3.2. Glycosylation Analysis of Synthetic Molecules from ZINC

The ZINC “for sale” subset is a vast collection of molecules that can be purchased; most of them are synthetic and non-tested for any activity in vitro or in vivo. In order to compare the glycosylation in NPs to the glycosylation of synthetic molecules, a structurally diverse set of 475,958 synthetic molecules was compiled from the ZINC15 “for sale” catalogue and analysed (Table 3). Only 851 (0.18%) molecules containing sugar moieties of any type were found. Among them, 39.25% (334) have linear sugar moieties, and 61.46% (523) have circular sugar moieties, with six molecules having both. A total of 0.04% (213) of molecules in this dataset are composed only of sugar units (Figure 17c).

Among the molecules containing circular sugars, 80.88% (423) have terminal circular glycosidic moieties (Figure 17a), and 19.12% (100) non-terminal ones. In the case of linear sugars, 89.82% (300) of linear sugar-containing molecules have terminal linear sugar moieties, and 10.18% (34) non-terminal ones. No molecules containing simultaneously terminal and non-terminal moieties were detected.

The curation of the dataset consisted of removing all the stereoisomers of the molecules from the ZINC “biogenic” subset and from COCONUT, and then grouping stereoisomers of the remaining synthetic molecules to avoid redundancy. Interestingly, on average 1.015 stereoisomers per normalised structure were included in the ZINC “for sale” dataset for every synthetic compound without glycosidic moieties. For sugar-containing synthetic molecules, on the other hand, there were 1.195 stereoisomers on average in the non-curated dataset.

### 3.3. Glycosylation Analysis of “Active In Vitro” Molecules from ZINC

Next, to assess the importance of sugars for bioactivity, a set of a priori synthetic molecules that are active in vitro was compiled from the ZINC15 “in vitro” catalogue. In this set of 65,620 molecules, 4.53% (2974) contain sugar moieties. Of these, 90.79% (2700) contain circular glycosides, 10.76% (320) linear ones, and 46 molecules have both. A total of 0.87% (572) of these active synthetic molecules are basically sugars (Figure 17c,d).

A total of 83.63% (2258) of circular sugar-containing molecules have terminal moieties of this type (Figure 17a), 18.70% (505) have non-terminal ones, and 2.33% (63) have both. Of linear sugar-containing molecules, 70.31% (225) contain terminal linear glycosidic moieties (Figure 17b), and 29.69% (95) non-terminal ones. No molecules were detected that contain both types of linear sugars. Considering stereoisomers that were filtered at curation, there were 1.348 stereoisomers per normalised synthetic molecular structure without glycosidic moieties in the original ZINC “in vitro” subset on average and 1.736 for every sugar-containing synthetic structure. Analysing the complete “in vitro” subset, i.e., without removal of biogenic molecules and their stereoisomers, 12,732 (6.98%) out of 182,514 molecules were found to contain sugar moieties, 94.59% (12,043) of which have circular glycosidic moieties, and 6.60% (840) linear ones. A total of 1087 molecules (0.60%) in the complete “in vitro” subset consist only of sugar units with side chains smaller than five heavy atoms.

## 4. Discussion

Sugars are known to enhance the bioactivities of NPs in numerous cases. They are, however, often overlooked in computational approaches to analyse candidates for drugs and other industrial applications of NPs and in the in silico generation of NP-like molecules.

In the NPs from the COCONUT dataset, around 12% of molecules have glycosidic moieties, and most of these are circular sugars. One can debate on the true nature of the difference between circular and linear sugars, as historically this distinction was not properly made: both representations of sugar-like structures were used, at least in the most common cases, such as glucose or fructose, whose linear and circular forms are regarded as equivalents. However, the COCONUT database contains 586 molecules with both linear and circular sugars in their structure, which shows the potential complexity of grasping the true glycon structure, especially while collecting it from the literature and diverse databases. Only a few (1404) molecules in the NP collection consist exclusively of various sugar moieties. As the latter tend to be monotonous, redundant structures, despite their potential almost infinite combination possibilities, they tend not to be represented in databases that much. Additionally, another possible bias can play a role in the glycosylation statistics of NP databases: historically, compound structure elucidation often used to be performed on aglycons to avoid monotonous sugar signals in the analysed spectra. Therefore, the number of glycosylated molecules among NPs in general can be considered as initially underestimated based on these databases.

Almost 75% of NPs with glycosides have them only in terminal position (i.e., their removal does not create two or more disconnected aglycon structures), which is unsurprising, as glycosylation is often a post-synthesis modification, aiming to “decorate” the molecule to change its properties [4,35]. The presence of non-terminal glycosides in 11,812 NPs indicates that these were likely added to the molecule during its synthesis, and probably do not impact the glycon activity in the same way that terminal glycosylation does. Additionally, these non-terminal sugar structures are unlikely to be the result of a canonical glycosylation event, i.e., by glycosylases or glycosyltransferases. Almost half of the glycosylated NPs have only one sugar moiety. It has been observed [4] that the number of glycosides on the aglycon can significantly impact its activity (e.g., more sugars favour a stronger binding to the target of the molecule), but often one is enough to ensure the desired activity.

As the results of review of glycosidic moieties in bacterial NPs by Elshahawi et al. [23] who analysed for occurrences in COCONUT show, the structural space of biogenic carbohydrates and carbohydrate moieties is large and diverse. The SRU with its default configuration is able to detect a large proportion of this diversity. Since these parameters are designed to identify redundant structures potentially obstructing analysis of the NP aglycon for in silico removal, they are deliberately limited to not include the whole diversity of glycosidic structures. These limits and what lies beyond them can be seen, for example, in the analysis of manually compiled bacterial NP sugar moieties (Figure 12, Figure 13, Figure 14, Figure 15 and Figure 16) and in the distribution of exocyclic oxygen ratio values of candidate sugar cycles in COCONUT (Figure 10 and Figure 11). However, it was also shown that one important feature of the SRU is its configurability, which allows the adaption of it for various applications and queries. Additionally, nearly all structures reported in the review dataset can, with adapted parameters, be identified as sugar moieties by the SRU. Consequently, these examples raise the question of what should be defined as a “true” sugar moiety that was likely added during a glycosylation event, and what would rather be a “sugar-like” moiety that more likely comes from a metabolic pathway unrelated to carbohydrates. This question may be subject to further investigations since the results of this study do not give indications for its answer but rather chart the space that has to be categorised.

A structurally diverse subset of synthetic molecules from ZINC “for sale” was also deglycosylated and analysed, in order to estimate glycosylation levels of a priori synthetic molecules and confirm once more that the presence of a sugar in the molecular structure is a property of natural or NP-derived molecules. Indeed, only 0.18% of these synthetic molecules have sugar moieties, with a completely different ratio of linear and circular sugars (respectively, 40% and 60%) from NPs (4% to 98%). Based on the results reported above, sugar moieties appear to be significantly less prevalent in synthetic molecules in general. The ZINC15 “active in vitro” collection of molecules is composed of molecules of various origins that have been tested in vitro and showed bioactivity, in its broad definition. Interestingly, 78% of this dataset corresponds to stereoisomers of NPs and biogenic molecules, which are not present in biological databases.

When considering the whole “in vitro” dataset, proportions of glycosylated molecules are comparably close to those in COCONUT. After removing the biogenic and NP stereoisomers, the proportions of active glycosylated molecules were much lower but still higher than in ZINC “for sale” (4.53% compared to 0.18%). Comparing the glycosylation of these three datasets, COCONUT, which is composed of NPs; ZINC “active in vitro”, composed of bioactive synthetic molecules; and ZINC “for sale”, made up of synthetic molecules, it is interesting to observe a certain correlation between bioactivity and the presence of sugar moieties in molecular structures.

The analysis of glycosylation of molecules of various origins showed that NPs contain a significantly higher proportion of sugar moieties than other molecule types and that glycosylation rates are higher in bioactive molecules. Therefore, SRU and large NP databases, such as COCONUT, should be combined for the discovery and prioritisation of bioactive molecules that may have pharmacological interest. It also might be interesting to formally include glycosylation into the NP-like and drug-like molecules generation, as it is already the case in some in vitro studies [36].

## 5. Conclusions

Glycosidic moieties attached to natural products serve multiple purposes, such as increasing solubility and changing the bioactivity and bioavailability of their aglycons [35].

The present study presents an overview and a statistical analysis of glycosidic moieties in natural products compiled in COCONUT, the largest open NP database. Here, about 12% of compounds have sugar moieties. Most of them are circular but linear sugar moieties can also be detected in 4% of the analysed NPs. Only 0.4% can be characterised as pure carbohydrates, which indicates that these are underrepresented in common NP databases. Another main finding of this study is that the potential space of glycosidic units in NPs is very diverse, and some structural features by which it can be characterised are presented. Comparatively, and maybe unsurprisingly, collections of synthetic molecules have significantly less glycosylated molecules: synthetic molecules from ZINC “for sale” have less than 0.02% of compounds with sugars, and a collection of synthetic active compounds from ZINC “in vitro” have only 4.5% of these. In turn, this indicates that glycosylated compounds are more prevalent among molecules with in vitro activity.

The analyses presented in this work offer a rapid overview of glycosylation in NP, which is valuable for various application fields, such as phytochemistry, and is underrepresented in the literature. To conclude, the SRU tool has been successfully applied to carry out a comprehensive glycosylation analysis of the COCONUT database, the largest open NP collection at the time of writing, and may serve as a useful tool for researchers in similar projects.

## Figures and Tables

**Figure 1 biomolecules-11-00486-f001:**
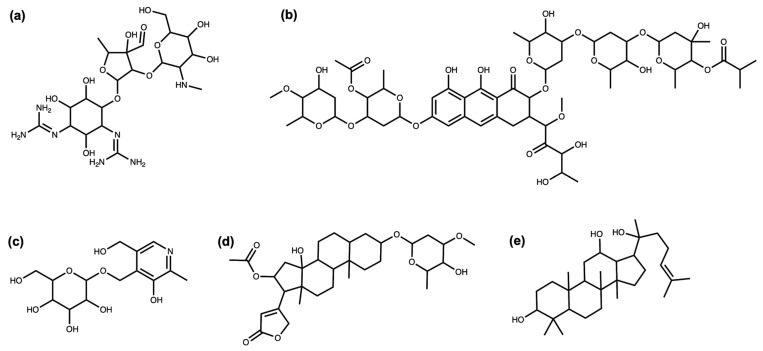
(**a**) Streptomycin. (**b**) Olivomycin A. (**c**) 4′-O-(beta-D-Glucopyranosyl)pyridoxine. (**d**) Oleandrin. (**e**) (20S)-Protopanaxadiol.

**Figure 2 biomolecules-11-00486-f002:**
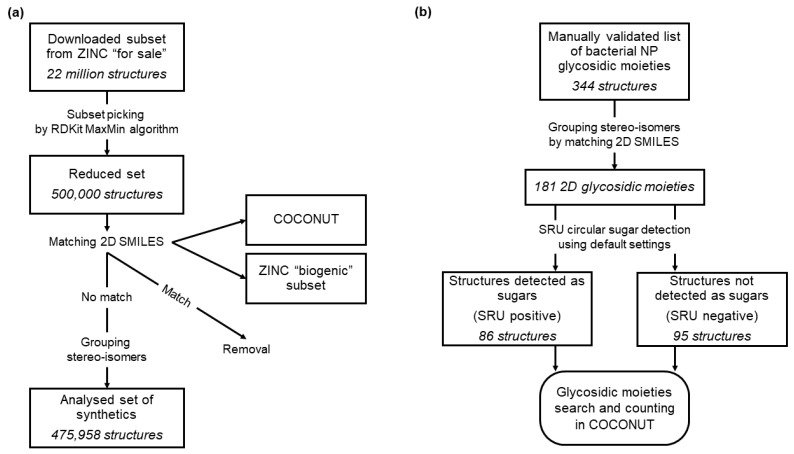
Data curation workflows. (**a**) Curation of synthetic molecules dataset from ZINC “for sale”. A total of 22 million molecules were downloaded from the ZINC “for sale” dataset and a diverse subset of 500,000 structures curated using the RDKit MaxMin algorithm. In the second step, all molecules matching structures in the ZINC “biogenic” subset or COCONUT were removed and stereo-isomers grouped. (**b**) Curation of glycosidic moieties of bacterial NP dataset. The manually curated dataset of glycosidic moieties of bacterial natural products (NPs) was first grouped for stereoisomers, resulting in 181 distinct structures. These were then categorised into two sets based on their detectability using the Sugar Removal Utility (SRU) default settings. With both sets, a substructure search in COCONUT was performed.

**Figure 3 biomolecules-11-00486-f003:**
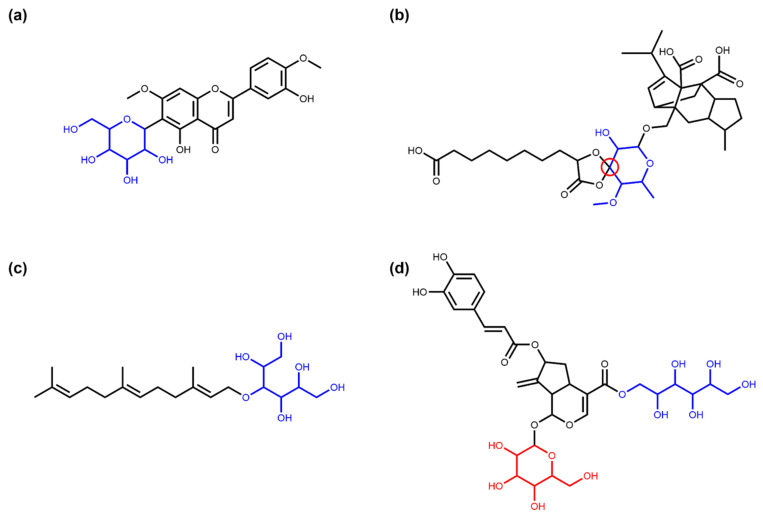
Examples of NPs with circular and linear sugar moieties. (**a**) 6-C-Glucopyranosylpilloin (COCONUT ID CNP0256263), a flavonoid structure with a cyclic sugar moiety (in blue) attached [24]. (**b**) Moriniafungin F (CNP0174070), a sordarin derivate where the sugar cycle (in blue) forms a spiro ring system with the adjacent cycle (spiro atom marked in red) [25]. (**c**) Cosmosporaside A (CNP0357595), a sesquiterpene glycoside with a linear sugar moiety (in blue) [26]. (**d**) Lavaudioside B (CNP0083726), an iridoid glycoside having both a linear (in blue) and a circular (in red) glycosidic moiety [27].

**Figure 4 biomolecules-11-00486-f004:**
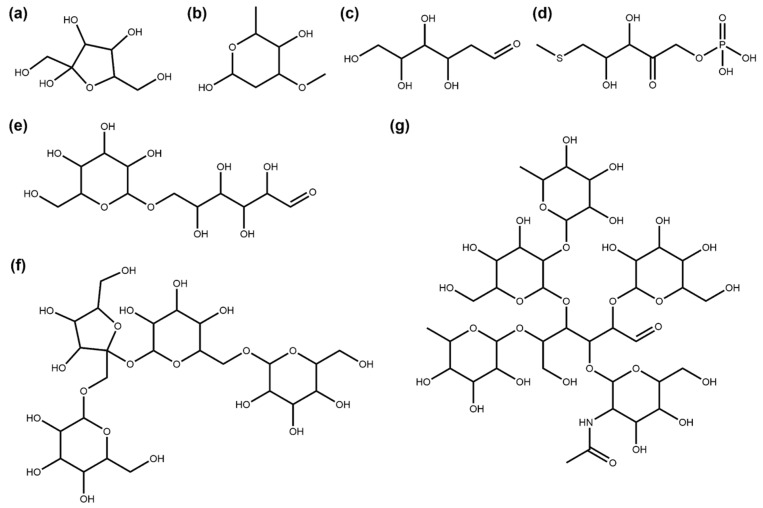
Examples of NPs that consist only of glycosidic moieties with side chains smaller than five heavy atoms. (**a**) Fructose (COCONUT ID CNP0232181), a commonly known monosaccharide. (**b**) Oleandrose (CNP0112845), a monosaccharide detected by the SRU despite its methyl and methoxy substituents and one deoxidised ring position. (**c**) 2-Deoxy-D-Glucose (CNP0019597), a linear monosaccharide. (**d**) S-Methyl-5-Thio-D-Ribulose-1-Phosphate (CNP0066170), a highly derivatised ribulose structure that the SRU still detects as a single linear sugar. (**e**) Isomaltose (CNP0226411), an isomer of maltose represented here with one glucose unit in open-chain formation which leads to recognition by the SRU as a polymer of one circular and one linear sugar moiety. (**f**) Lychnose (CNP0261531), a tetrasaccharide found in plants [28]. (**g**) Lacto-N-difucohexaose (CNP0084068), an oligosaccharide made up of five circular sugars and a central linear sugar that is found in human milk [29].

**Figure 5 biomolecules-11-00486-f005:**
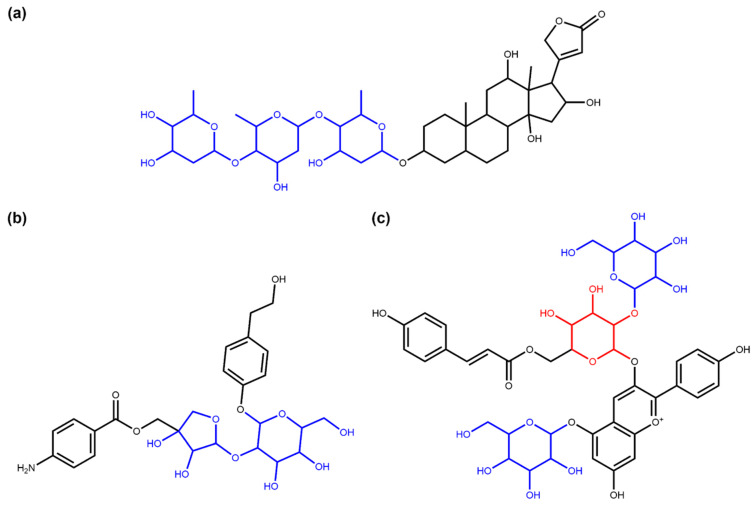
Examples of the terminal and non-terminal circular sugars identified in COCONUT NP. (**a**) Diginatin (COCONUT ID CNP0242430), a cardiac glycoside that has three terminal circular sugar moieties (in blue) (**b**) Cucurbitoside I (CNP0225316), a phenolic glycoside with two non-terminal circular sugars in its structure (in blue). (**c**) Pelargonidin-3-(2-Glucosyl-6-P-Coumarylglucoside)-5-Glucoside (CNP0028705), an anthocyanin [30] that contains both terminal (in blue) and non-terminal (in red) circular glycosidic moieties.

**Figure 6 biomolecules-11-00486-f006:**
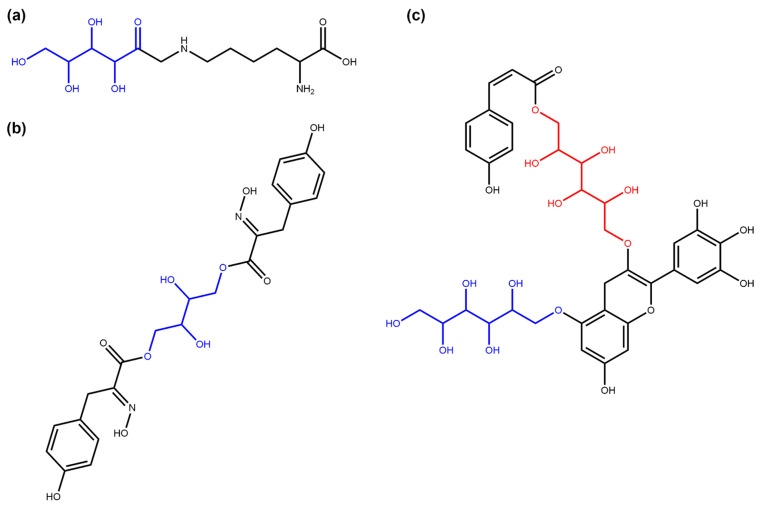
Examples of the terminal and non-terminal linear sugars identified in COCONUT NP. (**a**) Fructosyllysine (COCONUT ID CNP0082897), a terminal linear fructosyl moiety (in blue) attached to lysine. (**b**) Aspergillusol A (CNP0290091), a tyrosine-derived metabolite with a central non-terminal linear sugar (tetraol, in blue) structure [31]. (**c**) COCONUT NP CNP0321545, one of the two NPs in COCONUT with both terminal (in blue) and non-terminal (in red) linear sugar moieties in their structure.

**Figure 7 biomolecules-11-00486-f007:**
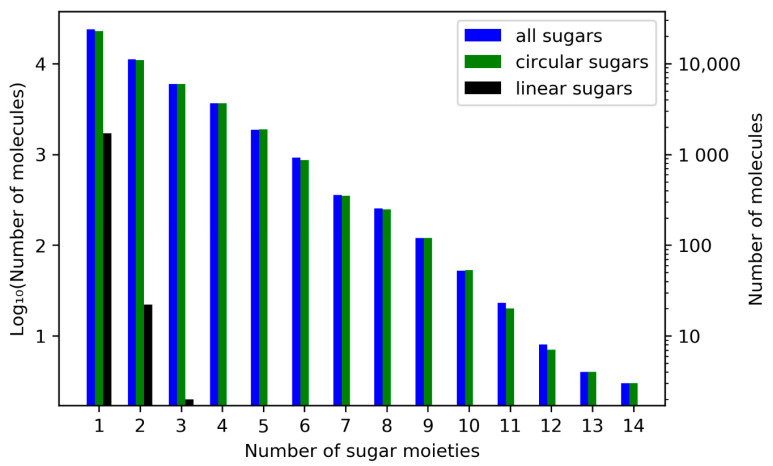
Frequencies for the total number of sugar moieties in a single molecule out of all molecules in COCONUT with glycosidic moieties. Numbers of molecules with the respective total number of sugar moieties are in a logarithmic scale on both ordinates. On the left vertical axis, units of the base-10 logarithm are plotted, and the untransformed numbers are given on the right. The blue bars disregard the types of sugars, whereas the other bars give the respective numbers only for circular (green) or linear (black) sugar moieties.

**Figure 8 biomolecules-11-00486-f008:**
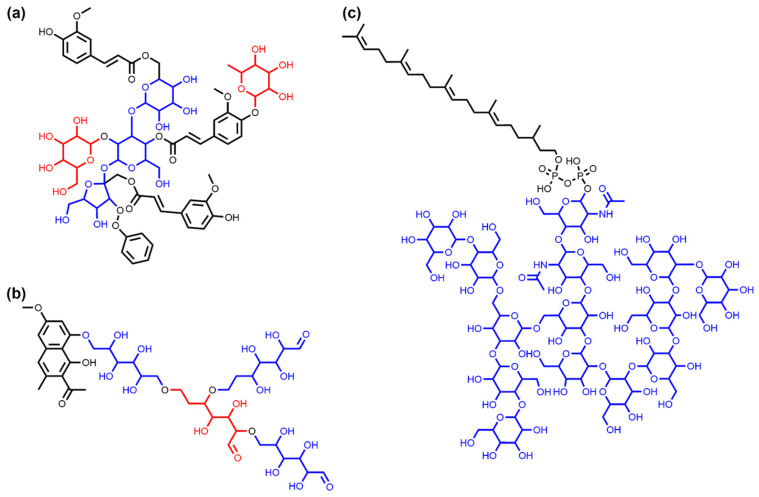
Examples of NPs with several circular or linear sugar moieties. (**a**) Dalmaisiose P (COCONUT ID CNP0319224), an oligosaccharide with five terminal (in red) and non-terminal (in blue) circular glycosidic moieties. (**b**) COCONUT molecule CNP0309951, the NPs in COCONUT with the most identified linear sugar moieties (coloured in red and blue only to mark the four distinct moieties, all of them are terminal). (**c**) Glycan G00008 (CNP0083402), a dolichyl diphosphooligosaccharide that is one of three NPs in COCONUT with the most (14 in total) identified circular sugar moieties.

**Figure 9 biomolecules-11-00486-f009:**
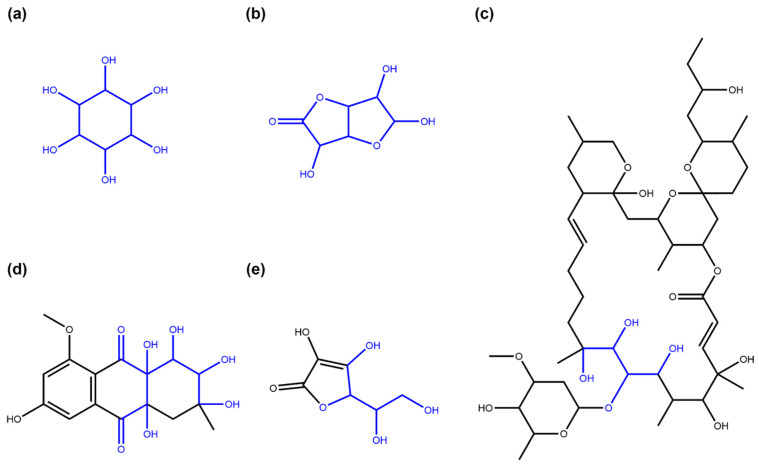
Examples of NP structures the SRU detects as linear sugars inside ring systems. (**a**) Inositol (COCONUT ID CNP0254016), a sugar-like cycle without ring oxygen (pseudo sugar) that is detected by the SRU linear sugar detection as a whole if the detection of linear sugars in rings is turned on. (**b**) 3,5,6-trihydroxy-hexahydrofuro[3,2-b]furan-2-one (CNP0261065), a structure that resembles two fused furanose rings and is detected as a whole by the linear sugar detection algorithm of the SRU if the option to allow detection of cyclic linear sugars is set. (**c**) Cytovaricin (CNP0392685), a macrolide that contains a linear sugar detectable by the SRU (in blue). (**d**) Chrysolandol (CNP0278335), a ring system with many hydroxy and keto groups that can be detected as a cyclic linear sugar (in blue). (**e**) Vitamin C (CNP0367758), a furanose-like structure that is undetectable for the circular sugar detection algorithm of the SRU because of its cyclic double-bond but can partwise be detected in linear sugar detection as a cyclic linear glycoside (detected structure in blue).

**Figure 10 biomolecules-11-00486-f010:**
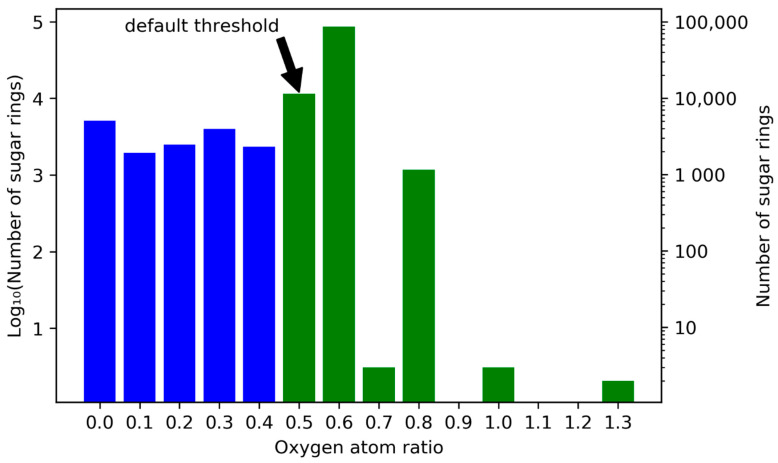
Frequencies for exocyclic oxygen atoms to atoms in ring ratio *r_exo_* for all cyclic sugar and sugar-like moieties in COCONUT. Plotted on the horizontal axis are intervals of exocyclic oxygen ratios that range from the respective denoted lower boundary to the value denoted for the following bar, i.e., [*r_exo,n_*, *r_exo,n+1_*). Numbers of sugar units located in the respective ratio intervals are in log scale on both ordinates. On the left vertical axis, units of the base-10 logarithm are plotted and the untransformed numbers are given on the right. The blue bars represent sugar rings that are excluded from sugar detection for having too few exocyclic oxygen atoms using the SRU default threshold. The green bars represent sugar moieties detectable with the default setting.

**Figure 11 biomolecules-11-00486-f011:**
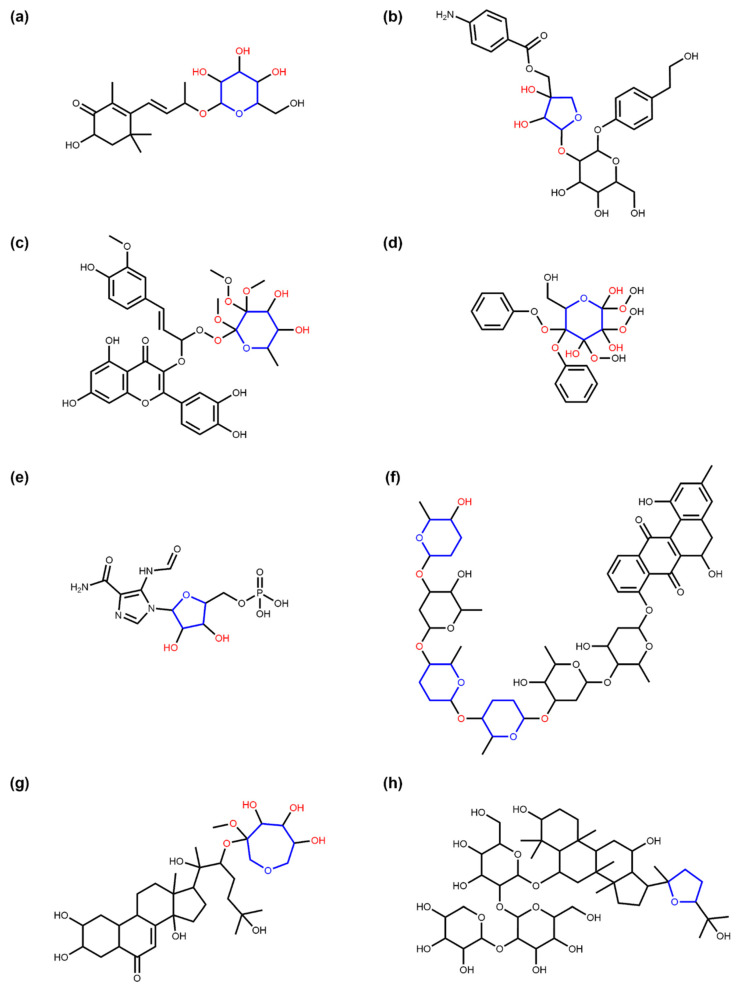
Examples of NP in COCONUT having circular sugar moieties with specific exocyclic oxygen atoms to atoms in ring ratio *r_exo_* values. The ratio is calculated by dividing the number of oxygen atoms attached to a furanose/pyranose/heptose ring *n_exo_* by the number of heavy atoms (C and O) in the ring *n_inc_* (see Equation (1)). In all subfigures, the respective sugar ring is marked in blue and all directly connected exocyclic oxygen atoms are marked in red. (**a**) Staphylionoside A (COCONUT ID CNP0250344), an NP with a pyranose sugar moiety that has an exocyclic oxygen ratio of 0.6 due to four connected oxygen atoms. (**b**) Cucurbitoside I (CNP0225316) contains a non-terminal furanose moiety with an exocyclic oxygen ratio of 0.6 due to the three connected oxygen atoms. (**c**) COCONUT NP CNP0030043 has a terminal pyranose moiety with an exocyclic oxygen ratio of 1.0 because the number of exocyclic oxygen atoms directly attached to the sugar ring equals the number of heavy atoms in it. (**d**) COCONUT NP CNP0075162 consists of two phenol moieties attached to a pyranose sugar that has an exocyclic oxygen ratio of 1.3 because there are eight oxygen atoms attached to it. Note that the interval [1.3, 1.4) contains only pyranoses. (**e**) Faicar (CNP0315195), an NP structure containing a furanose ring with an exocyclic oxygen ratio of 0.4 (two exocyclic oxygen atoms), a value that can only be reached by furanoses and heptoses. Moieties such as this can only be detected by the SRU if the exocyclic oxygen ratio threshold is lowered from the default value of 0.5. (**f**) Landomycin X (CNP0351566) has multiple terminal circular sugar-like moieties, of which three have an exocyclic oxygen ratio of only 0.3 (two attached oxygen atoms on a pyranose ring). If the SRU default options are used (exocyclic oxygen ratio threshold of 0.5 and no removal of non-terminal sugar moieties), the other three pyranoses with three exocyclic oxygen atoms, respectively, are detected as non-terminal sugar moieties and nothing is removed. By lowering the exocyclic oxygen ratio threshold to 0.3, all six pyranoses are detected as sugars and removed. A ratio of 0.3 can only be reached by pyranoses. (**g**) COCONUT NP CNP0306356 has an attached heptose moiety with five exocyclic oxygen atoms, and therefore an exocyclic oxygen ratio of 0.7, a value that can only be reached by heptoses. (**h**) Yesanchinoside C (CNP0107919) contains, in addition to three pyranose sugar moieties, a terminal furanose ring without any directly attached oxygen atoms. This ring, therefore, has an exocyclic oxygen ratio of 0 and can only be detected by the SRU if the respective threshold setting is lowered to this value, i.e., completely disregarding the number of attached oxygen atoms at the identification of glycosidic moieties.

**Figure 12 biomolecules-11-00486-f012:**
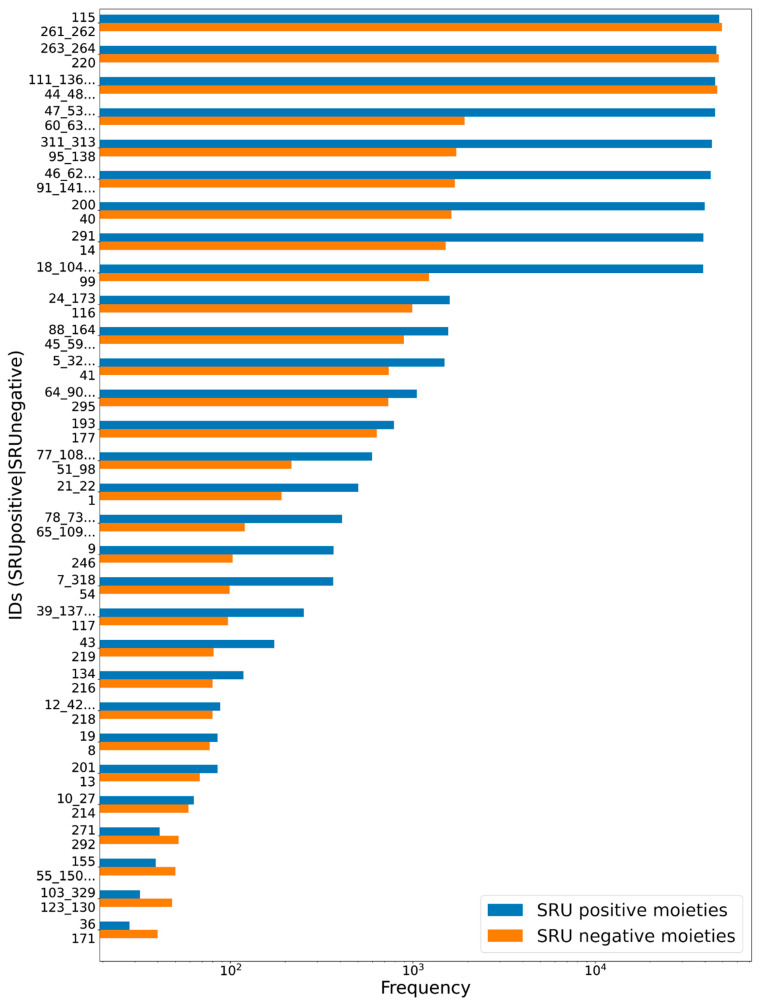
Frequencies of glycosidic moieties reported in bacterial NPs in COCONUT, grouped for detectability by the SRU using default settings. Sugar moieties detectable by the SRU are denoted “SRU positive moieties” (blue bars), and the remaining are denoted “SRU negative moieties” (orange bars). The ID on the vertical axis refers to the identifiers used in the original publication of the dataset, concatenated where stereoisomers were grouped. Frequencies, as detected by substructure matching, are in log-scale. Only moieties with at least ten matches in COCONUT were considered, and out of these the 30 most frequent are listed, respectively.

**Figure 13 biomolecules-11-00486-f013:**
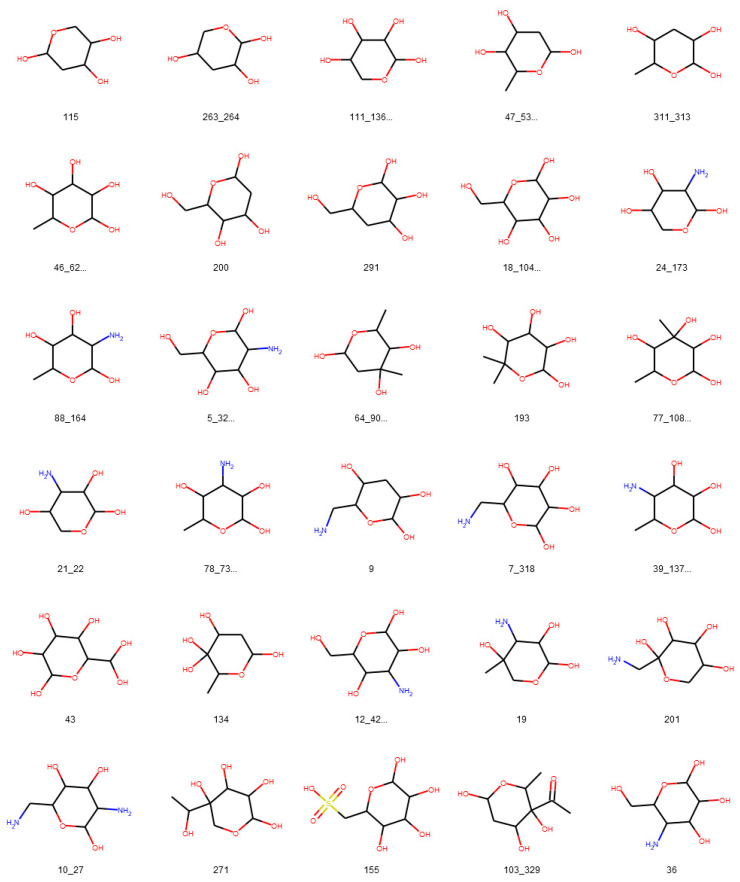
Glycosidic moieties reported in bacterial NPs detected at least ten times in COCONUT and detectable with the default SRU settings. All moieties are arranged for decreasing frequency in COCONUT, corresponding to the blue bars in Figure 12. The ID below the structures refers to the identifiers used in the original publication of the dataset, concatenated where stereoisomers were grouped.

**Figure 14 biomolecules-11-00486-f014:**
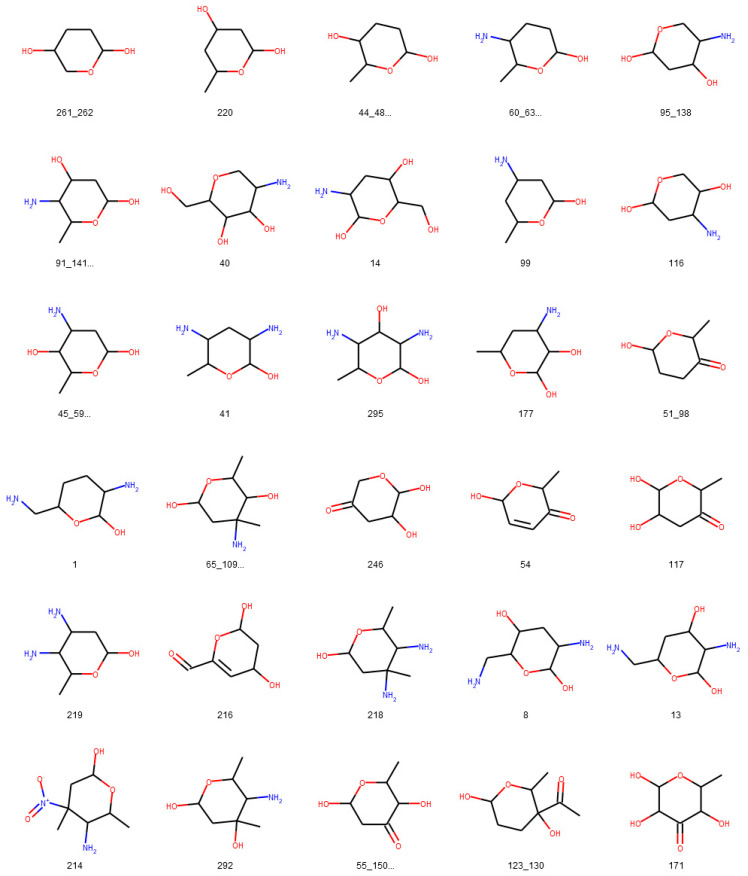
Glycosidic moieties reported in bacterial NPs detected at least ten times in COCONUT, not detectable with the default SRU settings. All moieties are arranged for decreasing frequency in COCONUT, corresponding to the orange bars in Figure 12. The ID below the structures refers to the identifiers used in the original publication of the dataset, concatenated where stereoisomers were grouped.

**Figure 15 biomolecules-11-00486-f015:**
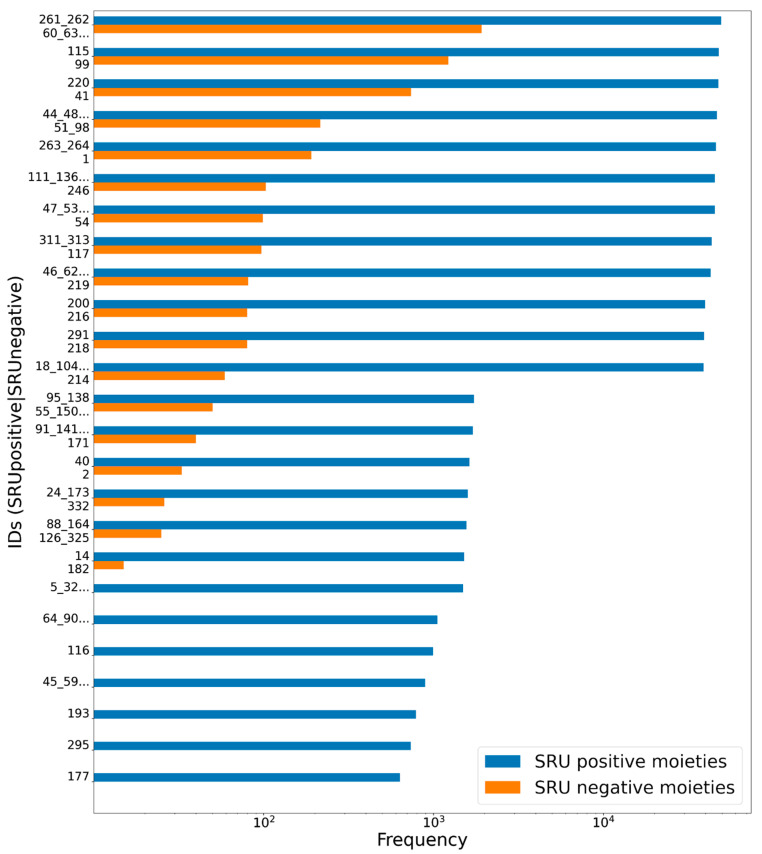
Frequencies of glycosidic moieties reported in bacterial NPs in COCONUT, grouped for detectability by the SRU after lowering the exocyclic oxygen ratio threshold setting to 0.3. Sugar moieties detectable by the SRU are denoted “SRU positive moieties” (blue bars), and the remaining are denoted “SRU negative moieties” (orange bars). The ID on the vertical axis refers to the identifiers used in the original publication of the dataset, concatenated where stereoisomers were grouped. Frequencies, as detected by substructure matching, are in log-scale. Only moieties with at least ten matches in COCONUT were considered and out of these, the 25 most frequent are listed, respectively.

**Figure 16 biomolecules-11-00486-f016:**
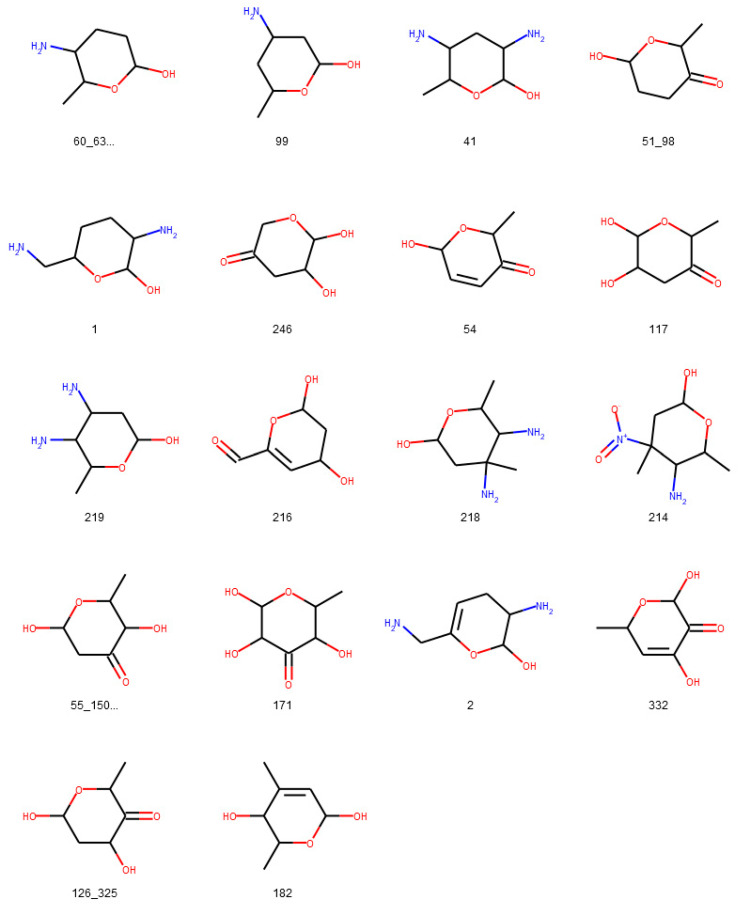
Glycosidic moieties reported in bacterial NPs detected at least ten times in COCONUT that cannot be detected by the SRU after lowering the exocyclic oxygen ratio threshold setting to 0.3. All moieties are arranged for decreasing frequency in COCONUT, corresponding to the orange bars in Figure 15. The ID below the structures refers to the identifiers used in the original publication of the dataset, concatenated where stereoisomers were grouped.

**Figure 17 biomolecules-11-00486-f017:**
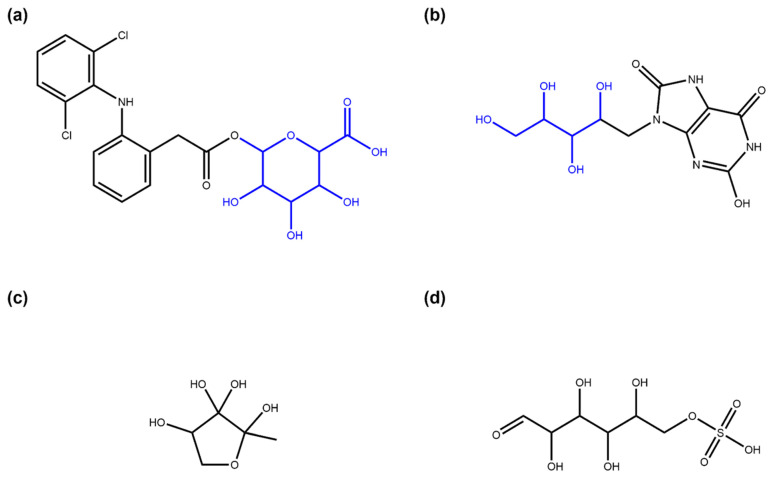
Examples of glycosidic molecules in ZINC “for sale” and “in vitro” subsets. (**a**) Diclofenac acyl glucuronide (ZINC ID ZINC35048352, part of the “for sale” and “in vitro” subsets), a metabolite of diclofenac that is suspected to mediate toxic side-effects of this drug [32]. The SRU detects the glucuronic acid moiety (in blue) added in metabolisation as a terminal circular sugar moiety. (**b**) 9-D-ribityl-1,3,7-trihydro-2,6,8-purinetrione (ZINC58650741, part of the “in vitro” subset), a uric acid derivative with a D-ribityl moiety (in blue, identified by the SRU) that can be used as a mimic of uridine diphosphate to inhibit human blood group B galactosyltransferase [33]. (**c**) (2R, 4S)-2-methyl-2,3,3,4-tetrahydroxytetrahydrofuran (ZINC6491067, part of the “for sale” and “in vitro” subsets) is not annotated in ZINC as NPs or metabolite and is also not part of COCONUT, although it has been identified as an autoinducer, a molecule used by bacteria for intercellular communication [34]. The SRU detects this structure as a circular monosaccharide. (**d**) D-glucose 6-sulfate (ZINC5132028, part of the “in vitro” subset), a sulfate analogue to the general metabolite D-glucose 6-phosphate in the open-chain form that is listed in ZINC as being found active in in vitro binding assays. The structure is recognised by the SRU as a linear monosaccharide.

**Table 1 biomolecules-11-00486-t001:** Proportions and numbers of the different detected types of glycosidic NPs in COCONUT.

	No. of Molecules	%
NPs in COCONUT	401,624	100.00%
Has sugars	48,118	11.98%
Has circular sugars	46,979	11.70%
Has only circular sugars	46,393	11.55%
Has linear sugars	1725	0.43%
Has only linear sugars	1139	0.28%
Has circular and linear sugars	586	0.15%
Consists only of sugars	1404	0.35%
Sugar monomer	425	0.11%
Circular sugar monomer	216	0.05%
Linear sugar monomer	209	0.05%
Sugar polymer	979	0.24%
Has terminal circular sugars	39,217	9.76%
Has only terminal circular sugars	35,167	8.76%
Has non-terminal circular sugars	11,182	2.78%
Has only non-terminal circular sugars	7762	1.93%
Has terminal and non-terminal circular sugars	4050	1.01%
Has terminal linear sugars	1119	0.28%
Has only terminal linear sugars	1117	0.28%
Has non-terminal linear sugars	608	0.15%
Has only non-terminal linear sugars	606	0.15%
Has terminal and non-terminal linear sugars	2	0.00%

**Table 2 biomolecules-11-00486-t002:** Proportions of different types of glycosidic moieties detected in COCONUT.

Type of Glycosidic Moieties	No. of Glycosidic Moieties	%
In COCONUT in total	99,943	100.00%
Circular	98,189	98.24%
Circular, terminal	80,927	80.97%
Circular, terminal, O-glycosidic bond	78,755	78.80%
Linear	1754	1.76%
Linear, terminal	1139	1.14%
Terminal	82,066	82.11%
Non-terminal	17,877	17.89%
Circular, furanose	4036	4.04%
Circular, pyranose	94,149	94.20%
Circular, heptose	4	0.00%
Linear, tetrose	686	0.69%
Linear, pentose	322	0.32%
Linear, hexose	719	0.72%
Linear, heptose	27	0.03%
Linear, ring substructure	11,950	11.96%

**Table 3 biomolecules-11-00486-t003:** Summary of glycosylation analysis results on ZINC datasets and comparison with COCONUT.

	Number of Molecules in							
	COCONUT	%	ZINC Synthetics “For Sale”	%	ZINC Synthetics “In Vitro”	%	ZINC “In Vitro”	%
Total molecules	401,624	100.00%	475,958	100.00%	65,620	100.00%	182,514	100.00%
Has sugars	48,118	11.98%	851	0.18%	2974	4.53%	12,732	6.98%
Has circular sugars	46,979	11.70%	523	0.11%	2700	4.11%	12,043	6.60%
Has linear sugars	1725	0.43%	334	0.07%	320	0.49%	840	0.46%
Has circular and linear sugars	586	0.15%	6	0.00%	46	0.07%	151	0.08%
Consists only of sugars	1404	0.35%	213	0.04%	572	0.87%	1087	0.60%
Has terminal circular sugars	39,217	9.76%	423	0.09%	2258	3.44%	9796	5.37%
Has non-terminal circular sugars	11,812	2.94%	100	0.02%	505	0.77%	2561	1.40%
Has terminal and non-terminal circular sugars	4050	1.01%	0	0.00%	63	0.10%	314	0.17%
Has terminal linear sugars	1119	0.28%	300	0.06%	225	0.34%	608	0.33%
Has non-terminal linear sugars	608	0.15%	34	0.01%	95	0.14%	232	0.13%
Has terminal and non-terminal linear sugars	2	0.00%	0	0.00%	0	0.00%	0	0.00%

## Data Availability

All data used in this work are open and accessible to everyone.

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
