# Peer review of "Description and Analysis of Glycosidic Residues in the Largest Open Natural Products Database"

_biomolecules, 2021, doi:10.3390/biom11040486_

Round 1

Reviewer 1 Report

Comments:

Dear Editor: 
The submitted manuscript describes assessing the description and analysis of glycosidic residues in the largest open natural products database. In this manuscript, the authors well studied the report and describe the presence of circular, linear, terminal, and non-terminal glycosidic units in NP, together with their importance in drug discovery.  Overall the manuscript is written well and a very useful topic also, an interesting topic for the readers. Therefore, the current version of the manuscript could be suitable for Biomolecules.

Author Response

We thank the reviewer for their encouraging comments.

Reviewer 2 Report

The reviewed paper is comprehensive statistical analysis concerning glycosidic residues in natural products developed based on COCONUT database. It highlights the significant role of sugar moieties as an active constituents of NP. The manuscript is well and clearly written. 
I found only a few editorial errors: 
- Line 307, 340-349: formatting of the text is erroneous; 
- references should be formatting according to Instruction for Authors 
- double dot in line 142

Author Response

We would like to thank the reviewer for their comments. Accordingly, the following changes have been made in the manuscript:

  • The double dot at line 142 (159 in the updated manuscript) has been corrected.
  • Text at lines 307 (341 in the updated manuscript) and 340-349 (375-383 in the updated manuscript) have been reformatted to fit the document style.
  • The reference list has been reformatted using Zotero and the MDPI references style.

Reviewer 3 Report

This systematic study of the glycosilation of natural products (NPs) represented in both public domain and commercial databases addreses an often neglected aspect of NPs.

The tools and approaches here described are adequate for the purpose of the work.

The information presented is valuable for phytochemists as it offers an 'at a glance' overview of both the variety and % of the occurrence of the different moieties present in glycosylated NPs , an aspect that is not present in most phytochemistry literature.

There are some minor issues with some terms used by the authors such as:

"Cardiac glycosides (Figure 1d), a well-known family of NP from plants, are cardiac steroids": perhaps you mean "Cardiac glycosides (Figure 1d), a well-known family of NP from plants, are cardiOTOXIC steroids "? Otherwise, it seems to imply they are naturally present in cardiac tissues!

Please recheck the manuscript for instances such as the one above mentioned. 

I am happy to recommend its publication 

Author Response

We would like to thank the reviewer for their comments. Concerning the cardiac steroids, we have reformulated the sentence in question (lines 56 to 58 in the updated manuscript) to hopefully make the origin and meaning of the term “cardiac steroids” more clear. It is not supposed to imply a natural presence of these compounds in cardiac tissues but highlights that these steroids are structurally similar to the adrenal cortical hormones and therefore potentially active on the same receptors which can have a beneficial effect.

Reviewer 4 Report

The subject is very interesting and few studies are in literature, but improvements are needed:

In the Introduction the context should be better described

The importance of database should be marked and related references inserted such as:

From Plant Compounds to Botanicals and Back: A Current Snapshot. doi: 10.3390/molecules23081844. 

The Database COCONUT should be introduced

The aim of work should be rewritten

More details in Methods including major details in Figure 2. The subpatagraph 2.1, 2.2 and 2.3 should be greatly implemented. The choice of COCONUT as optimal database should be justified and ZINC.

In Results, different paragraph and subparagraphs should be better linked.

Table 1 and Figures 3-7 should be better described and discussed in the text.

Lines 229-252 should be better explained.

Figure 10 should be better discussed in the text.

3.1.2 should be implemented in the description

Limits, advantages, practical applications and novelty character should be marked.

Author Response

We would like to thank the reviewer for their comments. Here is the point by point response:

 “In the Introduction the context should be better described [...] The aim of work should be rewritten”

-> The introduction has been extended (lines 76 to 80) to better describe the context and aim of the study.

“The importance of database should be marked and related references inserted such as:

From Plant Compounds to Botanicals and Back: A Current Snapshot. doi: 10.3390/molecules23081844. The Database COCONUT should be introduced [...] The choice of COCONUT as the optimal database should be justified and ZINC.”

-> We thank the reviewer for the suggestion. The manuscript has been extended with more extended explanations about the COCONUT (lines 102-108) and the ZINC (lines 111-115) databases and the reasons for their use. However, after a careful examination of the suggested reference to be inserted we didn't judge it to be pertinent for the current manuscript.

“More details in Methods including major details in Figure 2. The subpatagraph 2.1, 2.2 and 2.3 should be greatly implemented.”

-> In Figure 2, more details have been added and it has been extended to also illustrate the curation of the used ZINC datasets.

“In Results, different paragraph and subparagraphs should be better linked.”

-> Some paragraphs in the Results section have been re-written to address this issue.

“Table 1 and Figures 3-7 should be better described and discussed in the text.”

-> The legend of Table 1 has been improved. We would like to point out that lines 165-227 describe and discuss the results presented in Table 1. Concerning the figures, they are now addressed more explicitly in the text.

“Lines 229-252 should be better explained.”

-> The subparagraph (lines 253-285 in the updated manuscript) now has a more detailed introduction and a statement about the level of detail of part of the presented results was added at the end.

“Figure 10 should be better discussed in the text.”

-> Figure 10 has an extensive legend and is being discussed in the text at lines 304-326. Also, we realized that in the discussion we quoted erroneously a different figure, which has been updated at lines 512-513.

“3.1.2 should be implemented in the description”

-> With the updated figure numbers, the results section 3.1.2, on the bacterial NP glycosidic moieties, is being discussed in the "Discussion" section in lines 504 to 521.

“Limits, advantages, practical applications and novelty character should be marked.”

-> Both the end of the introduction and the discussion have been extended to clarify these points and address these suggestions.

Round 2

Reviewer 4 Report

The paper is suitable for publication